# How Does it Sound? Generation of Rhythmic Soundtracks for Human Movement Videos

**Kun Su** [*†]          **Xiulong Liu** [*†]          **Eli Shlizerman** [‡§]

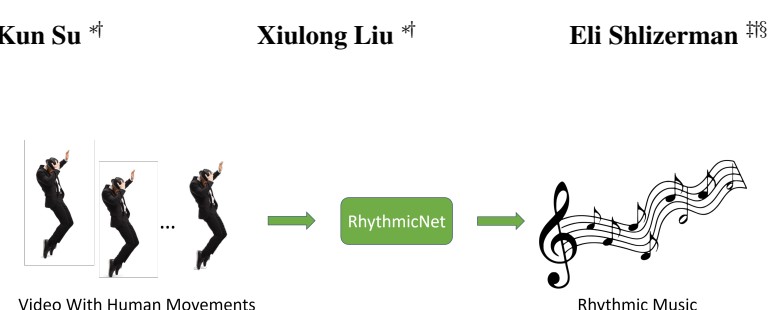

Figure 1: RhythmicNet: Given an input of a silent human movement video, RhythmicNet generates a soundtrack for it.

## Abstract

One of the primary purposes of video is to capture people and their unique activities. It is often the case that the experience of watching the video can be enhanced by adding a musical soundtrack that is in-sync with the rhythmic features of these activities. How would this soundtrack sound? Such a problem is challenging since little is known about capturing the rhythmic nature of free body movements. In this work, we explore this problem and propose a novel system, called 'RhythmicNet', which takes as an input a video with human movements and generates a soundtrack for it. RhythmicNet works directly with human movements, by extracting skeleton keypoints and implementing a sequence of models translating them to rhythmic sounds. RhythmicNet follows the natural process of music improvisation which includes the prescription of streams of the beat, the rhythm and the melody. In particular, RhythmicNet first infers the music beat and the style pattern from body keypoints per each frame to produce the rhythm. Next, it implements a transformer-based model to generate the hits of drum instruments and implements a U-net based model to generate the velocity and the offsets of the instruments. Additional types of instruments are added to the soundtrack by further conditioning on generated drum sounds. We evaluate RhythmicNet on large scale video datasets that include body movements with inherit sound association, such as dance, as well as 'in the wild' internet videos of various movements and actions. We show that the method can generate plausible music that aligns with different types of human movements.

## 1   Introduction

Rhythmic sounds are everywhere, from raindrops falling on surfaces, to birds chirping, to machines generating unique sound patterns. When sounds accompany visual scenes, they enhance the perception of the scene by complementing it with additional cues such as semantic association of events, means of communication, drawing attention to parts of the scene, and many more. For visual scenes

---
[*]These authors contributed equally.

[†]Department of Electrical & Computer Engineering, University of Washington, Seattle, USA.

[‡]Department of Applied Mathematics, University of Washington, Seattle, USA

[§]Corresponding author: shlizee@uw.edu

35th Conference on Neural Information Processing Systems (NeurIPS 2021).

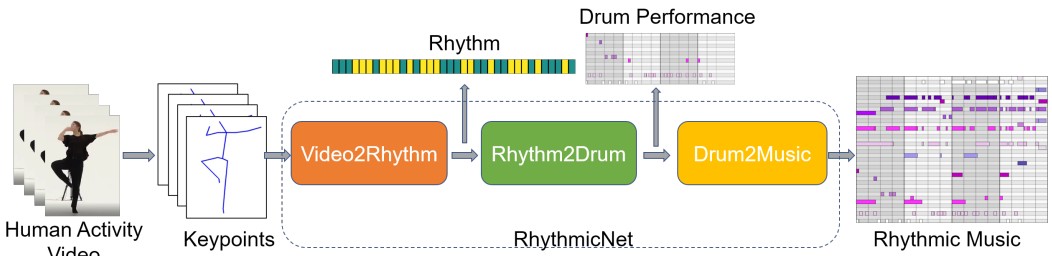

Figure 2: System Overview of RhythmicNet. Keypoints are extracted from human activity video and are processed through Video2Rhythm stage to generate the rhythm. Afterwards Rhythm2Drum converts the rhythm to drum performance. In the last step, Drum2Music component adds additional instrument tracks on top of the drum track.

that include activity of people, rhythmical music that is in-sync with the rhythm of body movements can emphasize the actions of the person and enhance the perception of the activity [1, 2]. Indeed, to support such synchrony, a usual practice is that a musical soundtrack is chosen manually in professionally edited videos.

Drum instruments serve as the fundamental part in music by generating the underlying leading rhythm patterns. While drum instruments vary in shape, form, and mechanics, their main purpose is to set the essential rhythm for any music. Indeed, drums are known to have existed from around 6000 BC, and even beforehand there were instruments based on principle of hitting two objects and generating sounds [3]. On top of drum patterns, additional instruments add secondary patterns and melody, creating rich multifaceted music. In modern music, in composition and improvisation, it is also the case that composers would start a new musical piece by designing the rhythm for the corresponding drum track. As the piece evolves, additional accompanying instruments tracks are gradually superimposed on top of the drum track to produce the final music.

Inspired by the possibility of associating rhythmic soundtracks to videos, in this work we explore automatic generation of rhythmic music correlated with human body movements. We follow similar music composition and improvisation steps as in music improvisation by first generating the rhythm of the music that is strongly correlated with the beat and movements patterns. Such rhythm can then be then used to generate novel drums music accompanying the body movements. With the rhythm being inferred, we follow further steps of music improvisation and add new instruments (piano and guitar) tracks to enrich the music. In summary, we address the challenge of generating a rhythmic soundtrack for a human movement video by proposing a novel pipeline named 'RhythmicNet', which translates human movements from the domain of video to rhythmic music with three sequential components: Video2Rhythm, Rhythm2Drum, and Drum2Music.

In the first stage of RhythmicNet, given a human movement video, we extract the keypoints from the video and use a spatio-temporal graph convolutional network [4] in conjunction with transformer encoder [5] to capture motion features for estimation of music beats. Since music beats are periodic and there are various visual changes occurring in human movements, we propose an additional stream, called the style, which captures fast movements. The combination of the two streams constitutes the movements rhythm and guides music generation in the next stage, called Rhythm2Drum. This stage includes an encoder-decoder transformer that given the rhythm, generates the drums performance hits and a U-net [6] which subsequently generates drums velocities and offsets. We find that these two stages are critical for generation of quality drum music. In the last stage, called Drum2Music, we complete the drum music by adopting an encoder-decoder architecture using transformer-XL [7] to generate a music track of either piano or guitar conditioning on the generated drum performance. An overview of RhythmicNet is shown in Fig. 2. Our main contributions are: (i) To the best of our knowledge, we are the first to generate a novel musical soundtrack that is in-sync with human activities. (ii) We introduce an entire pipeline, named 'RhythmicNet', which implements three stages to complete the transformation. (iii) RhythmicNet is robust and generalizable. Experiments on datasets of large-scale dance videos and 'in the wild' internet videos show that music generated by RhythmicNet will be consistent with human body movements in videos.

## 2 Related Work

Generation of sounds for a video is a challenging problem since it aims to relate two signals that are indirectly correlated. It belongs to the class of problems of *Audio-Visual learning*, which deals with exploration and leveraging of the correlation of both audio and video for tasks such as audio-visual correspondence [8, 9, 10, 11], video sound separation [12, 13, 14, 15], audio-visual event localization [16], transformations of audio to body movements [17, 18, 19], lips movements [20] and talking faces [21, 22, 23]. Audio-visual systems are usually developed by using multi-modal learning techniques which have been shown effective in action recognition [24, 25], speech question answering [26, 27, 28, 29, 30], 3D world physical simulation [31], and medical images analysis [32, 33, 34, 35, 36].

Several approaches were proposed for the relation of sounds to a video. A deep learning approach showed the potential of such application by proposing a recurrent neural network to predict the audio features of impact sounds from videos. The approach was able to produce a waveform from these features [37]. In a subsequent work, a conditional generative adversarial network was proposed to achieve cross-modal audio-visual generation of musical performances [38]. In both methods, single image was used as an input, and the network performed supervision on instrument classes to generate a low-resolution spectrogram. Concurrently, for natural sounds, a Sample RNN-based method [39] has been introduced to generate sounds such as baby crying, water flowing, given a visual scene. This approach was enhanced by an audio forwarding regularizer that considers the real sound as an input and outputs bottle-necked sound features which provide stronger supervision for natural sound predictions only from visual features [40]. Compared to natural sounds with relatively simple characteristics, music contains more complex elements. While such problem is more challenging, the possibility to correlate movement and sounds was shown by a rule-based sensor system which succeeded to convert sensed motion to music notes [41].

In recent years there has been remarkable progress in the generation of music from video. An interactive background music synthesis algorithm guided by visual content was introduced to synthesize dynamic background music for different scenarios [42]. The method, however, relied on reference music retrieval and could not generate new music directly. Direct music generation approaches have been developed for videos capturing a musician playing an instrument. A ResNet-based method was proposed to predict the pitch and the onsets events, given video frames of top-view videos of pianists playing the piano [43]. Later, Audeo [44] demonstrated the possibility to transcribe video to high-quality music. While the results of such methods are promising, the generation is limited to a single instrument. Thereby, Foley Music [45] proposed a Graph-Transformer network to generate Midi events from body keypoints and achieved convincing synthesized music from Midi. Further, Multi-instrumentalist Net [46] showed generation of music waveform of different instruments in an unsupervised way. While these approaches demonstrate the possibilities of generating music from videos, the videos need to contain solid visual cues such as instruments to indicate the types of music being generated. It still remains unclear whether it is possible to generate music when such visual cues do not exist. With respect to human movement, this would be extracting the characteristics of the movement and attempting to match music with them. In this regard, a recent novel approach of dance beat tracking was proposed [47]. The approach is aimed at detecting the characteristics of musical beats from a video of a dance by using visual information only. Inspired by this work, we design a novel methodology to estimate in a precise way musical characteristics, such as beats, from movements and utilize them to improvise new music.

There has been vital recent progress in the generation of music from its representations, such as symbolic representations, as well. In particular, Musical Instrument Digital Interface (Midi) representation has been shown to be useful in modeling and generating music. Initial works converted Midi into piano-roll representation and used generative adversarial networks [48] or variational autoencoders [49, 50] to generate new music. A limitation of the piano-roll is that it may result in memory inefficiency when the length of the music is too long. In order to address this limitation, event-based representation has been proposed and was shown to be a useful and efficient representation in modeling music [51, 52, 53]. While the event-based representation enabled models to obtain convincing generated results, it lacks metrical structure, leading to unsteady beats in the generated samples. Thereby, recently, a new representation called Remi was proposed to impose a metrical structure in the input data so that the models can include awareness of the beat-bar-phrase hierarchical structure in the music [54]. In our work, we utilize the Remi representation by converting the Midi into

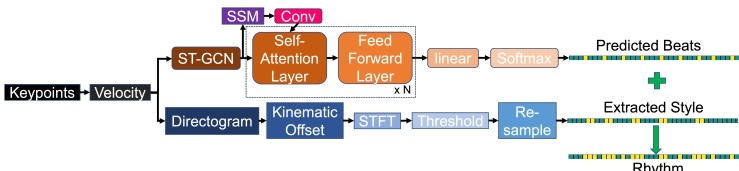

Figure 3: Detailed schematics of the components in the Video2Rhythm stage.

Remi in the Drum2Music stage. While the methods mentioned above generate unconditional music, it was shown to be possible to constrain music generation. For example, it was proposed to constrain generative models to sample for predefined attributes [55]. Systems such as Jukebox [56] and MuseNet [57] showed the possibility of generating music based on user preferences which correspond to network model specifically trained with labeled tokens as a conditioning input. Furthermore, a Transformer autoencoder has been proposed to aggregate encoding of Midi data across time to obtain a global representation of style from a given performance. Such a global representation can be used to control the style of the music [58]. Additional models have been proposed, such as a model capable of generating kick drums given conditional signals including beat, downbeat, onset of snare and Bass [59]. In RhythmicNet, conditioning additional music instruments on the drum track is expected to provide a richer soundtrack. For this purpose, in the Drum2music stage, we utilize the Transformer autoencoder and consider the drum track as the conditioning input and generate the track of another musical instrument, such as piano or guitar.

## 3 Methods

RhythmicNet includes three sequential components: 1) Association of *rhythm* with *human movements* (Video2Rhythm), 2) Generation of *drum track* from *rhythm* (Rhythm2Drum), 3) *Adding instruments* to the drum track (Drum2Music). We describe the details of each stage below.

***Video2Rhythm.*** We decompose the rhythm into two streams: beats and style. We propose a novel model to predict music beats and a kinematic offsets based approach to extract style patterns from human movements.

*Music Beats Prediction.* Beat is a binary periodic signal determined by fixed tempo, and it is obtained by music beat prediction network, which learns the beat by pairing body keypoints with ground truth music beats in a supervised way. To predict regular music beats from human body movements, we extract 2D skeleton keypoints via the OpenPose framework [60] and perform first-order difference to obtain the velocity for each video. Motion sequences is considered as three dimensional tensor $X \in \mathbb{R}^{V \times T \times 2}$ where $V$ is the number of keypoints, $T$ is the number of frames, and the last dimension indicates the 2D coordinates. We formulate the prediction of music beats as a temporal binary classification problem: Given the skeleton keypoints $X$, we aim to generate the output with the same length $Y \in \mathbb{R}^T$, where each frame is classified into 'beat' ($y = 1$) or 'non-beat' ($y = 0$).

We encode the keypoints using a spatio-temporal graph convolutional neural network (ST-GCN) [4]. Such encoding represents the skeleton sequence as an undirected graph $G = (V, E)$, where each node $v_i \in V$ corresponds to a key point of the human body and edges reflect the connectivity of body keypoints. The sequence passes through a spatial GCN to obtain the features at each frame independently, and then a temporal convolution is applied to the features to aggregate the temporal cues. The encoded motion features are then represented as $P = AXW_SW_T \in \mathbb{R}^{V \times T_v \times C_v}$, where $X$ is the input, $A \in \mathbb{R}^{V \times V}$ is the adjacency of matrix of the graph defined based on the body keypoints connections. $W_S$ and $W_T$ are the weight matrices of spatial graph convolution and temporal convolution. $T_v$ and $C_v$ indicate the number of temporal dimension and feature channels. We obtain the final motion features $P \in \mathbb{R}^{T_v \times C_v}$ by averaging the node features.

Given the motion feature $P$, we use a transformer encoder that contains a stack of multi-head self-attention layers to learn the correlation between different frames. Due to the periodicity of the music beats, we introduce two components to allow the model to capture them more accurately: 1) We adopt a relative position encoding [61] to allow attention to explicitly resolve the distance between two tokens in a sequence instead of using common positional sinusoids to represent timing information.

This encoding is critical for modeling the timing in music where relative differences matter more than their absolute values [52]. 2) We use temporal self-similarity matrix of motion features (SSM), which has been shown effective in human action recognition in regularization of the transformer and counting the repetitions of periodic movements [62, 63, 64]. SSM can be constructed by computing all pairwise similarities $S_{ij} = f(P_i, P_j)$ between pairs of frame-level motion features $P_i$ and $P_j$, where $f(\cdot)$ is the similarity function. We use the negative of the squared euclidean distance as the similarity function, $f(a, b) = -||a - b||^2$, followed by taking softmax over the time axis. SSM has only one channel and it goes through a convolution layer $\hat{S} = \mathrm{Conv}(S)$ and then added to every attention head in the self attention component implemented as

$$\mathrm{Attention}(Q, K, V) = \mathrm{Softmax}(\frac{QK^T + \hat{S} + R}{\sqrt{D_k}})V,$$

where $Q, K, V$ are the standard query, key and value respectively, and $R$ is the ordered relative position encoding for each possible pairwise distance among pairs of query and key on each head. We train the model using weighted binary cross-entropy loss that puts more weight toward the beat category to address imbalances.

In comparison with previous work [47], the combination of graph representation, relative self-attention and SSM components enables the model to better capture the spatial-temporal structures in body dynamics which allows for more accurate beat estimation.

The output of the network is the beat activation function; i.e., for each video frame, the model predicts its probability of being a 'beat' frame. To obtain beat positions, we apply an algorithm based on HMM decoding proposed in [65].

*Style Extraction.* While beats represent the monotonic periodic pattern occurring at fixed time intervals (i.e. periodic signal), there are additional a-periodic components in the rhythm. In particular, between two music beats, there are typically various irregular movements that contribute to the rhythm. In contrast to beats, these patterns are inconsistent and it is unclear how to systematically extract such patterns from visual information. We, therefore, define an additional stream, called style, which records incidences of transitional movements of the human body, such as rapid and sudden movements. For prediction of such events, we apply a rule-based approach since the definition of style is implicit and there is no data to learn a mapping from body keypoints to transitional movements. The style is defined as a binary stream that indicates transition time points as $1$ and non-transitional time points as $0$. We compose the style stream by implementing several steps based on spectral analysis of kinematic offsets of the motion [66]. The first step is to compute kinematic offsets. Kinematic offsets are 1D time series signal representing the average acceleration of the human body over time. To obtain kinematic offsets, we calculate the directogram of the motion by factoring it into different angles. Given $F_t(j, t)$ as the velocity magnitude of joint $j$ at time $t$, we formulate the directogram $D(t, \theta)$ [67] as:

$$D(t, \theta) = \sum_j F_t(j, t) \mathbb{1}_\theta(\angle F_t(j, t)), \text{ where } \mathbb{1}_\theta(\phi) = \begin{cases} 1 & |\theta - \phi| \leq 2\pi/N_{\mathrm{bins}} \\ 0 & \text{otherwise} \end{cases} \quad (1)$$

The indicator function $\mathbb{1}_\theta(\phi)$ is used to distribute the motion of all joints into $N_{bins}$ angular intervals. Then the first-order difference of the directogram is calculated to obtain the acceleration of motion across different angles. The mean acceleration in the positive direction measures motion strength (i.e., the larger the value, the more remarkable in motion strength) and corresponds to the kinematic offsets.

Once kinematic offsets are obtained, in the next step we perform a Short-Time-Fourier Transform (STFT) on them to identify peaks in the change of acceleration. The highest frequency bin in STFT (out of 8) represents the most profound transitions in the signal and we use the highest frequency bin to extract the style patterns from motion. The peaks are defined as 10% top magnitudes over the duration of the video. We mark the timepoints of the peaks as $1$ and other timepoints as $0$. Since STFT results with low temporal resolution (due to hop-size set to 4 for efficient computation) we upsample the binary signal by the hop size to obtain a binary signal that matches the resolution of the video. The output signal is re-sampled to have the same sampling rate as the music beats.

*Rhythm Composition.* We obtain the rhythm by adding the streams of the beats and the style into a single signal. The rhythm should correspond to the correlation of body movements with the tempo of the soundtrack.

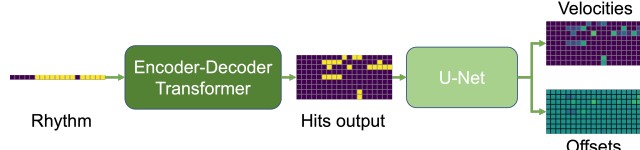

Figure 4: Detailed schematics of the components in the Rhythm2Drum stage.

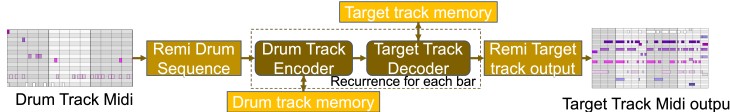

Figure 5: Detailed schematics of the components in the Drum2Music stage.

**Rhythm2Drum.** The stage of Rhythm2Drum interprets the provided rhythm from previous stage into drum sounds. In this stage we follow the GrooveVAE setup [50], where each drum track can be represented by three matrices: hits, velocities, and offsets. The hits represent the presence of drum onsets and is a binary matrix $H \in \mathbb{R}^{N \times T}$, where $N$ is the number of drum instruments and $T$ is the number of time steps (one per 16-th note). The velocities is a continuous matrix, $V$, that reflects how hard drums are struck, with values in the range of $[0, 1]$. The offsets $O$ is also a continuous matrix and stores the timing offsets, with values in the range of $[-0.5, 0.5]$. These values indicate how far and in which direction each note's timing lie, relative to the nearest 16-th note. The matrices $V$ $O$, and $H$ have the same shape.

Given the input rhythm sequence $Y \in \mathbb{R}^{1 \times T}$, we aim to generate the $H$, $V$ and $O$. In contrast to GrooveVAE [50], which models all three matrices simultaneously with multiple losses, we model $H$, $V$, and $O$ smoothly in two steps using the combination of an encoder-decoder transformer [5] and a U-net [6]. In the first step, the binary rhythm is passed as an input to the transformer encoder. In the decoder, the $H$ matrix is converted into word a sequence defined by a small vocabulary set of all possible combinations of hits, and is mapped back to a binary matrix for the final output. We observe that autoregressively learning the hits $H$ as a word sequence, the transformer can generate more natural and diverse drum onsets. We train the transformer with the cross-entropy loss. In the second step, we add style patterns (velocity and offsets) to the onsets. Since $H$ has the same shape as $V$ and $O$, we can consider it as a transformation between 2 images of the same shape. To achieve such transformation, we adopt a U-net [6] to take the onset matrix $H$ as an input and to generate $V$ and $O$. We use Mean-Square Error (MSE) loss for U-net optimization. Finally, we convert the generated matrices $H$, $V$ and $O$ to the Midi representation to produce the drum track.

**Drum2Music.** In this last stage we add further instruments to enrich the soundtrack. Since the drum track contains rhythmic music, we propose to condition the additional instrument stream on the generated drum track. Specifically, we propose an encoder-decoder architecture, such that the encoder receives the drum track as an input, and the decoder generates the track of another instrument. We consider the piano or guitar as the additional instruments, since these are dominant instruments. We use Remi representation [54] to represent multi-track music. Compared to the commonly-used Midi-like event representation [52], the Remi representation includes information such as Tempo changes, Chord, Position, and Bar, which allow our model to learn the dependency of note events occurring at same positions across bars. For both the encoder and decoder, we adopt the transformer-XL network model which extends the transformer by including the recurrence mechanism [7]. The recurrence mechanism enables the model to leverage the information of past tokens beyond the current training segment and to look further into the history.

The encoder contains stack of multi-head self-attention layers. Its output $E_i$ can be represented as: $E_i = \text{Enc}(x_i, M_i^E)$, where $M_i^E$ is the encoder memory used for the $i$-th bar input and the encoder hidden state sequence computed in previous recurrent steps. Similarly, in the decoder, the prediction of $j$-th token of the $i$-th bar $y_{i,j}$ is formulated as $y_{i,j} = \text{Dec}(y_{i,t<j}, M_i^D, E_i)$, where $y_{i,t<j}$ are the previously generated tokens in the same bar, $M_i^D$ is the decoder memory used for $i$-th bar, and $E_i$ is the corresponding encoder output of the same bar. The decoder consists of a stack of layers with casual self-attention, cross-attention to the encoder output, and feed-forward network.

| Models\Metrics | CML$_c$ (%) | CML$_t$ (%) | Cem (%) | F (%) |
|---|---|---|---|---|
| TCN [47] | 44.97 | 45.15 | 48.14 | 63.04 |
| TF | 16.07 | 16.24 | 32.85 | 46.90 |
| ST-GCN | 54.89 | 55.45 | 49.23 | 64.78 |
| ST-GCN+TF | 61.89 | 62.34 | 55.09 | 71.93 |
| ST-GCN+TF+SSM | 63.20 | 63.58 | 57.72 | 73.07 |
| ST-GCN+TF+RelAttn | 68.01 | 68.31 | 59.19 | 74.67 |
| **ST-GCN+TF+SSM+RelAttn** | **71.43** (+26.46%) | **71.94** (+26.79%) | **61.59** (+13.45%) | **75.79** (+12.75%) |

Table 1: Music beat prediction evaluation. The abbreviation of each component stands for: TF (transformer), ST-GCN (spatio-temporal graph convolutional network), SSM (Self-similarity Matrix), RelAttn (Relative Attention); F (F-score measure), Cem (Cemgil's score), CML$_c$ (Correct metrical level continuous accuracy), CML$_t$ (Correct metrical level total accuracy). Bold font indicates the best value.

| Model\Metric (lower better) | NDB | MSE Velocity | MSE Offsets |
|---|---|---|---|
| GrooveVAE [50] | 46 | 0.0437 | 0.0402 |
| TF multi-outputs w.o. hits sequence | 44 | 0.0507 | 0.0348 |
| TF multi-outputs w. hits sequence | 39 | 0.0493 | 0.0369 |
| **TF w. hits sequence + Unet** | **39** (↓15%) | **0.0267** (↓40%) | **0.0169** (↓58%) |

Table 2: Rhythm2Drum performance evaluation. Abbreviations stand for: TF (encoder-decoder transformer), Multi-outputs (Predict the hits, velocities and offsets simultaneously), w./w.o. hits sequence (whether using word tokens to represent the hits). Bold font indicates the best value.

For the training data, we split the music piece into segments with a total number of bars. In the encoder, for recurrent step $i$, we provide the $i$-th bar of drum performance $x_i$ to the transformer-XL. We adopt a teacher forcing strategy and feed the ground truth tokens into the decoder to generate the next tokens. We minimize the negative log-likelihood (NLL) between generated tokens and ground truth tokens to optimize the model. During inference, the drum track is given to the encoder for each bar and the tokens in the decoder are generated one by one. Finally, we use the temperature-controlled stochastic top-k sampling method [68] to randomly generate a new music track.

## 4  Experiments & Results

*Datasets.* We use the AIST Dance Video Database, a large-scale collection of dance videos in 60fps for training and testing of Video2Rhythm [69]. This database includes 10 street dance genres, 35 dancers, 9 camera viewpoints, and 60 musical pieces covering 12 types of tempo. For each genre, we use 1080 dance videos, resulting in total of 10, 800 videos. We split the samples into train/validate/test sets by 0.8/0.1/0.1 based on the dance genres, dancers, and camera ids.

For Rhythm2Drum, we use the Groove Midi dataset [50] which contains 1150 Midi files and over 22, 000 measures of drumming. We split the data into 0.8/0.1/0.1 of train/validate/test sets.

For Drum2Music, we extract two subsets of Lakh Midi dataset [70] to separately train Drum2Piano and Drum2Guitar models. For Drum2Piano, we select the Midi files that contain both tracks of drums and acoustic piano with at least 16 bars, and we consider 16 bars to be a single segment. This results in 34991/1944/1944 segments for train/validate/test sets respectively. For drum2guitar, we perform a similar selection to obtain 12904/717/717 segments for train/validate/test sets respectively.

*Implementation Details.* We use Pytorch [71] to implement all models in RhymicNet with two Titan X GPUs. For all videos, we extract 17 keypoints of body joints. In Video2Rhythm, the network contains a 10-layer ST-GCN and a 2-layer transformer encoder with 2-head attention. For the style extraction part, the motion sequence is down-sampled to 15fps to calculate the kinematic offsets. The number of bins used for the directogram is 12, and a 16-point FFT with hop-size of 4 is applied to extract candidate styles. Each detected style is repeated 4 times to match the hop size, and then is up-sampled to original time resolution of 60fps and re-indexed in unit of quarter note based on estimated tempo. In Rhythm2Drum, the Hits transformer includes 3 layers, and 4 heads in both the encoder and the decoder. The vocabulary size of the decoder input is 152, consisting of all possible combinations of 9 types of drum hits in the dataset. U-net that generates velocity and offsets contains

| Metrics | PC/bar | PI | IOI | PCH ↑ | NLH ↑ | NLL ↓ |
|---|---|---|---|---|---|---|
| Dataset (Piano) | 5.48 | 6.16 | 0.31 | - | - | - |
| Drum2Piano w.o. memory | 7.17 | 4.63 | 0.12 | 0.63 | 0.52 | 0.77 |
| Drum2Piano | **6.82** | **5.86** | **0.14** | **0.63** | **0.54** | **0.53** |
| Dataset (Guitar) | 5.33 | 5.51 | 0.22 | - | - | - |
| Drum2Guitar w.o. memory | 3.54 | 8.94 | 0.52 | 0.56 | 0.46 | 0.58 |
| Drum2Guitar | **5.63** | **5.69** | **0.13** | **0.64** | **0.51** | **0.40** |

Table 3: Drum2Music evaluation. For PC/bar, PI, IOI values, the closer to the dataset the better. For PCH and NLH values, the larger, the better.

4 down-sample blocks with channel sizes of 16, 32, 64, 128. In Drum2Music, the model consists of a recurrent transformer encoder and a recurrent transformer decoder. We set the number of encoder layers, decoder layers, encoder heads and decoder heads to $4, 8, 8$, and $8$ respectively. The length of the training input tokens and the length of the memory is 256. We provide additional configuration details in the supplementary materials. *Code.* System setup and code are available in a Github repository[5].

*Video2Rhythm Evaluation.* Following the rubrics proposed for musical beat tracking [72], we compute the performance in terms of F-score measure, Cemgil's score (Cem), and Correct Metrical Level continuity required/not required ($CML_{c/t}$) score. To compare with existing approaches, we implement a baseline temporal convolutional network (TCN) for beat prediction [47]. The comparison and ablation results are shown in Table 1. The best method of Video2Rhythm (ST-GCN+TF+SSM+RelAttn) significantly outperforms the baseline model in all metrics by a large margin. In particular, the continuity scores outperform the baseline model by more than $25\%$, indicating the estimated beat sequence is significantly more consistent.

*Rhythm2Drum Evaluation.* We use several metrics to evaluate Rhythm2Drum. For measuring the diversity of the generated drum hits, we adopt the Number of Statistically-Different Bins (NDB) metric proposed and used in [73, 74, 45]. To compute NDB, we cluster all training examples into $k = 50$ Voronoi cells by K-means. The generated examples are then assigned to the nearest cell. NDB is reported as the number of cells where the number of training examples is statistically significantly different from the number of generated examples by a two-sample Binomial test. For each model, we generate 9000 samples from the testing set and perform the comparison. For evaluation of velocities and offsets, we compute the Mean-Squared Error (MSE) for the test set. We compare our methods with the baseline model GrooveVAE [50]. The results are shown in Table 2. The results show that using hits sequences to generate the drum track enables a more diverse set of samples such that the next U-net component, which generates velocities and offsets, in turn will generate more realistic drumming sounds.

*Drum2Music Evaluation.* To evaluate the generated piano and guitar tracks, we use objective metrics such as PC/bar (pitch count per bar), PI (average pitch interval), IOI (average inter-onset interval) described in [75]. For these metrics we compare the statistics calculated on the test dataset and on the generated music. For additional metrics of PCH (pitch class histogram) and NLH (note length histogram), we calculate the overlapping area (OA) between the statistics on the test dataset and the generated music for each sample and report the average of them. In addition, we compare the NLL loss based on the validation set. The numerical results are shown in Table 3. We compare two versions (with and without using memory) to show the effectiveness of the recurrence mechanism. Our results show that for both Drum2Piano and Drum2Guitar with recurrent encoder-decoder transformer (i.e. with memory), the NLL loss is lower for the validation set and the statistics of the generated samples are much closer to the test dataset than the no-memory counterpart.

*In the wild Experiments and Qualitative Evaluation.* In Fig. 6, we show a set of examples of generated soundtracks for video clips in AIST dataset and 'in the wild' clips downloaded from YouTube. For the AIST dataset, we generate and compare tracks of predicted beats with the ground-truth(GT) beats. The predicted and GT tracks appear to be in close agreement. We then demonstrate the extracted style and its correspondence to frames which exhibit special movements. The beat and style tracks constitute the rhythm from which the waveform of drum music is generated. To test and demonstrate the generality of RhythmicNet we apply it to videos clips of various human activity. Examples of such

---

[5]https://github.com/shlizee/RhythmicNet

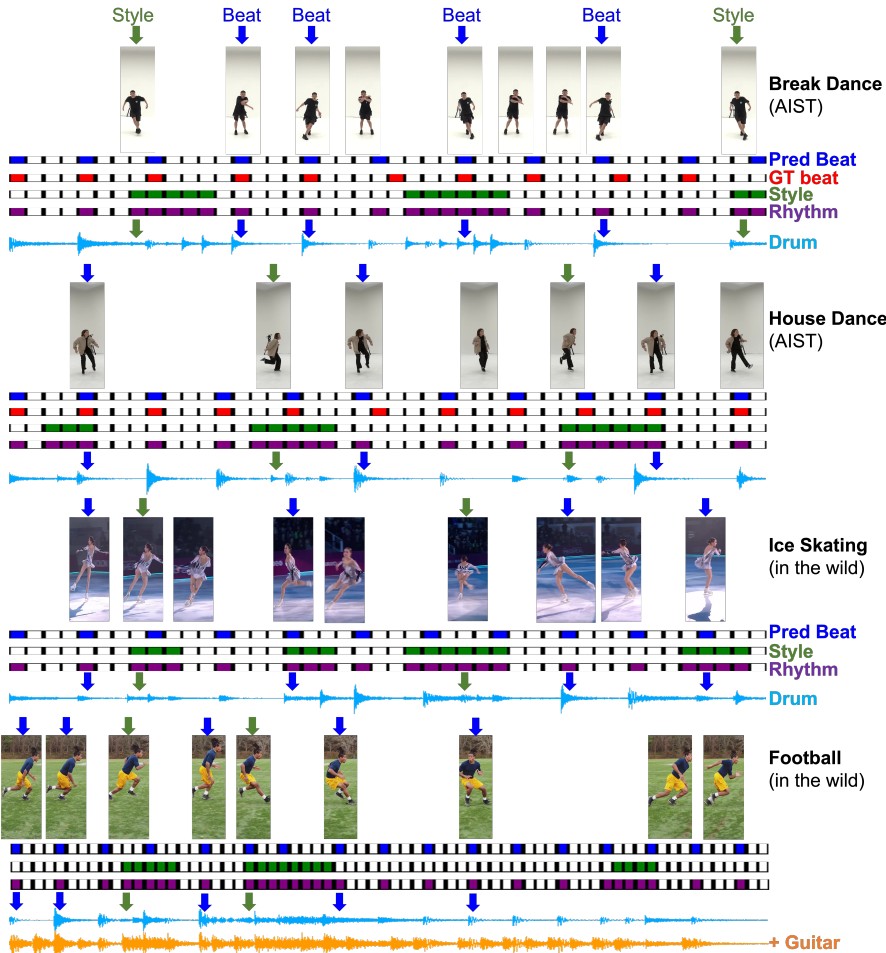

Figure 6: Examples of generated beat and style streams and corresponding audio waveforms for dance (AIST videos) and 'in the wild' videos. Dark Blue: predicted beats, Red: ground truth beats, Green: extracted style, Purple: rhythm, Light Blue: audio waveform of generated drums. Supplementary Materials include additional examples and sounded video clips.

activities are shown in Fig. 6 and include Ice Skating and Playing Football. We provide additional examples and videos clips along with the soundtracks in the Supplementary Materials. The generated rhythms are well synchronized with the videos and the drum track appears to be in-sync with the activities. In addition, we demonstrate the generated waveform an additional instrument (guitar) in Fig. 6. Additional instruments indeed provide a richer music that accompanies the movements.

*Human Perceptual Evaluation of Soundtrack Music.* In addition to the objective evaluation of the different components of RhythmicNet we also performed human perceptual surveys using Amazon Mechanical Turk. These surveys were intended to evaluate the effectiveness of RhythmicNet generated soundtracks to align with the movements and the extent that the generated soundtrack enhance the overall perception of the video compared with various soundtrack controls. Since RhythmicNet ultimately targets in the wild videos, for which there are no given background soundtracks we ran three surveys that focused on these videos. For all surveys, no background on the survey or RhythmicNet was given to the participants to avoid perceptual biases. We surveyed $85$ participants individually, where each participant was asked to evaluate 10 videos each with around $10$ seconds ($850$ segments in total) along with different generated soundtracks.

In the first survey, we asked people (non-experts) to choose the video that they prefer, including a video without soundtrack and 3 variations of soundtracks generated by our approach (drums-only or drums with another instrument). Results in Table 4 clearly indicate a preference of a video

|  | Soundtrack Preference | | | |
|---|---|---|---|---|
|  | No Soundtrack | Drums Only | Drums + Piano | Drums + Guitar |
| votes | 7.3% | 31.2% | 32.1% | 29.4% |

Table 4: Soundtrack preference.

| Soundtrack match to the video | | | Soundtrack match to the video (Ablation) | | |
|---|---|---|---|---|---|
| Random | Shuffle | RhythmicNet | Random + GrooVAE | Video2Rhythm + GrooVAE | Video2Rhythm + Rhythm2Drum |
| 30.8% | 27.8% | 41.4% | 23.3% | 33.3% | 43.4% |

Table 5: Soundtracks match to movements in the video.

with a soundtrack. Furthermore, interestingly, preference for which instruments are included in the track split almost equally between the 3 provided variations, with slight preference for tracks with Drums+Piano.

In the second survey, we asked people to watch the same human activity video with different soundtracks and answer the question: "In which video the sound best matches the movements?". The given options of the soundtracks were generated soundtracks with Random, Shuffle and RhythmicNet rhythms. The Random drum track was generated with *Rhythm2Drum* method with a random rhythm with 50% chance to be ON or OFF at each time step. The chance of 50% was chosen such that there is a significant probability that the a rhythm that sounds like a real rhythm will be sampled. We found that sampling with lower probability would generate rhythms that do not sound well at all. The Shuffle drum track was generated with Rhythm2Drum method but the order in the rhythm is shuffled. RhythmicNet option corresponded to the drum track generated with Rhythm2Drum method. From results shown in Table 5(left) we observe a clear indication that the drum tracks generated with our method are chosen to be the best match to the movements more frequently (41.4% (Ours) v.s. 30.8% (Random) and 27.8% (Shuffle)).

In an additional survey, we performed a perceptual ablation study to test how the two components, Video2Rhythm and Rhythm2Drum, influence the perception of the soundtrack compared to baseline approaches. Survey results shown in Table 5(right) and suggest that in comparison to the baseline these two components significantly improve the perception of the soundtrack.

## 5   Conclusion and Discussion

In this exploratory work, we have considered a creative task of automatically generating novel rhythmic soundtracks consistent with human body movements captured in a video. Our results show that RhythmicNet pipeline is able to achieve this creative task and generate soundtracks that align with movements and enhance the perception of them when the video is being watched. At its core, RhythmicNet defines and implements a systemic approach of soundtrack generation by following the process of music improvisation in which a rhythm of movements is established and is translated to drumming music with potentially additional accompanying instruments. We foresee future potential applications in video creation and editing, which RhythmicNet can pave the way to unlock. As features for music generation we have chosen body keypoints, while it is unclear which features are most informative for music generation. For human body movements, body keypoints are strongly correlated with movements, and in addition, body keypoints are efficiently computed per each frame. In terms of limitations and future enhancements of the current setup of RhythmicNet, novel components will need to be considered to address the transition from rhythmic drum track to a full bodied music with a symphony of instruments, since currently the addition of a single instrument (piano or guitar) to the drum track is implemented. Furthermore, enabling RhythmicNet to operate in real-time would allow the music to be interactive with people and their movements. However this may require a more computationally extensive generative approach. Due to the fact that the main cue for the generated music is human body movements, one possible concern could be that the soundtrack generated with RhythmicNet could be used to manipulate the original sounds of the video, and to create a fake impression of people activity. Also additional concern is that generated music could sound too familiar to the music on which the models in RhythmicNet have been trained. These are common concerns in the application of generative models. Failure in RhythmicNet may bring up an unsatisfying soundtrack but we do not expect serious consequences from this.

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

## Funding Disclosure

