# OpenReview forum: "How Does it Sound?"
_NeurIPS.cc/2021/Conference — NeurIPS 2021 Poster_

### Official Review · Reviewer_587L · 2021-07-13

**Rating:** 6
**Confidence:** 3

**Summary:**

The paper proposes a new method to produce a rhythmic sound consisting of basic instruments such as (piano roll, guitar chords and drum beat) given an input video containing at least one human activity. The authors leverage the input human action dynamics first using a keypoint estimation network and their velocity features. The latter features as used consequently to produce a rhythmic pattern which is used to generate a sequence of drum beats. The aforementioned drum beat sequence is then used as an output or as conditioning to a late-stage generation of a piano roll or a sequence of guitar chords. The intermediate representations and the architecture choices are discussed for each staged of the process. The authors provide both qualitative and quantitative results (opinion scores, video examples) for showing the effectiveness of their proposed architecture.

**Limitations And Societal Impact:**

The authors do vaguely discuss a few limitations of their work in the conclusion section but they can also include any of the points that I raised here (if they do not wish to address them in the rebuttal).

In my opinion, this work raises some potential negative societal impacts that the authors do not discuss. For instance, the generated music could be heavily biased against specific patterns, styles or artists as well as raising potential copyright issues (e.g. imagine that a specific generated pattern sounds extremely similar to a copyrighted song, who would get the responsibility?).

**Main Review:**

First of all, I would like to mention that this paper discusses a really novel idea of generating musical patterns from human action videos and the architectural choices, as well as the experiments are extremely well described. The main limitation of the paper are the results of the proposed method as well as the evaluation metrics described in the main paper which do not capture the great potential of this paper. Please see below a more detailed review:

To the best of my understanding, the paper is very original since I am not aware of almost any work which tried to generate a musical pattern solely from human actions pertaining to dance or other random movements. I really like the idea and the possible real-world applications that this paper enables. The paper also details many experimental design choices that researchers would need to replicate the results which is a strong plus. Moreover, the many examples provided in the supplementary material is also really important for the readers to understand the performance of the proposed model. However, I have the following concerns / questions about this work:
- I understand why the authors base their whole musical pattern generation architecture on top of the drum beat sequence generation because a fundamental aspect such as the base rhythm is a very effective way to build the generation upon (in parallel, multiple approaches in speech generation base their own approaches on first estimating pitch). However, the network could use this beat rhythm to do much more than using it to condition the generation of other basic instruments sequences. Specifically, the authors could use this rhythm as conditioning with other (probably pre-trained) networks such as Jukebox, etc. In this sense, the output could actually enjoy much richer musical textures and patterns. Have the authors tried something along those lines or they aim to do so in the future?
- Although the keypoints and their velocity are convenient for estimating a base rhythm (e.g. drum beat) and might contain some information for the style, I would expect a simple video encoder to be able to produce representations containing much richer conditional style-information. For example, I have seen multiple videos in the supplementary material where the generated musical patterns for a hip-hop dancer and a ballerina would have somewhat similar styles (although the beat was somewhat aligned). I would expect the authors to at least discuss this future direction.
- The authors can save space from Figures 3-5 since they are extremely sparse. Moreover, Figure 6 is nice but the authors can split the examples to 1-2 for the main text and put the rest of them in the supplementary material since they do not necessarily add up to the main point of the paper.
- A very important omission of the main paper is the display of the qualitative assessment of the generated videos. To the best of my understanding, such an assessment is really important for generation tasks and especially for so subjective tasks as the one addressed in this paper. There is only a small evaluation like this in the supplementary material where the only question asked was about the synchronization. I do not criticize that those results are not perfect and the proposed model is compared to almost random baselines but they are very important to exist in the main paper. I would also suggest to the authors to include a subjective evaluation for the actual preference of humans of the produced musical patterns.
- Another main problem is that there are many statements in the paper that build a really high excitement about the results which are far from perfect (e.g. Lines 20-21 “We show that the method can generate plausible music 21 that aligns well with different types of human movements.” Or Lines 66-67 “the output music generated by RhythmicNet will be consistent with the human body movements in 67 videos in large-scale dance dataset and ‘in the wild’ internet videos.”). I checked almost all of the examples provided in the supplementary material but I could easily imagine one piano roll being replaced to another video, probably showing that this first approach to the problem (still significant though) does not respect the style and of course leaving a large performance gap for future works to build on. I would strongly suggest to the authors to be more explicit with their wording if they decide to not include a subjective evaluation of the preference of the generated musical pattern over other baselines (e.g. Line 313 “soundtracks consistent with human body movements” → “drum beats and piano rolls synchronized with human body movements”).


**Time Spent Reviewing:**

4 hours

---

> ### Author Response · Authors · 2021-08-10
> **Reply to Reviewer 587L**
>
> We thank the reviewer for a thoughtful review, valuable feedback and acknowledgement of the novelty and interest in our work. We address the questions raised in the review below.
>
> - Indeed, the rhythm could be generated in various ways and with other instruments than drums directly.  We identified the drums as the most direct and intuitive approach to achieve an association of sounds with human movements, especially in in-the-wild videos with various movements. Additional approaches could be interesting future potential directions. Networks like Jukebox could be applicable, however we found that it may require significant implicit engineering setup and extremely computationally expensive training to map rhythm to general music. Pretrained networks for such a task do not exist.
>
> - A video encoder using various features could indeed be an extension of our work for association of video scenes and audio. However this may require a more computationally extensive generation model and it is unclear which features are most informative for music generation. For human body movements, body keypoints are strongly correlated with movements  which supports the goal of our work to associate movements with music. Furthermore, body keypoints could be efficiently computed per each frame. We will add such discussion to the paper.
>
> - As recommended by the reviewer, we will provide additional detail on the methodology and will revise Figures 3-5 and move some of the examples to Supplementary Materials. With available space, we intend to reorganize the methods section and clearly define and explain the style extraction component and add the perceptual surveys as described below.
>
> - We thank the reviewer for suggesting to clarify the subjective evaluation of the results especially in terms of the overall evaluation of the videos with soundtracks. Since our application is ultimately targeted toward videos for which there is no given background soundtrack (in the wild videos) we ran two additional surveys that focus on these videos and they will be placed in the final version of the paper.
>   **Soundtrack Preference (Total surveyed: 850)**
> |       | No Soundtrack | Drums Only  | Drums+Piano | Drums+Guitar |
> |-------|---------------|-------------|-------------|--------------|
> | Votes | 7.3% (62)     | 31.2% (265) | 32.1% (273) | 29.4% (250)  |
>
>   In the first survey, we asked people to choose the video that they prefer, including a video without a soundtrack and 3 variations of soundtracks that our approach generates (drums-only or drums with another instrument). Results clearly indicate a clear preference for a video with a soundtrack and interestingly preference for which track splits almost equally with slight preference for tracks with drums and piano.
>    **Soundtrack match to the video (ablations) (Total Count: 850)**
> |       | Random Rhythm + Baseline Drum Generation (GrooVAE) | Video2Rhythm (Ours)  + Baseline Drum Generation (GrooVAE) | Video2Rhythm (Ours)  +  Rhythm2Drum (Ours) |
> |-------|----------------------------------------------------|-----------------------------------------------------------|--------------------------------------------|
> | Votes | 23.3% (198)                                        | 33.3% (283)                                               | 43.4% (369)                                |
>
>   In another survey, we performed a subjective perceptual ablation study to test how our two components Video2Rhythm and Rhythm2Drum influence the perception of the soundtrack compared to baseline approaches. Our results are shown above and suggest that in comparison to the baseline these two components significantly improve the perception of the soundtrack. As the reviewer recommends, we will place these results along with discussion and the surveys reported, in the main paper.
>
> - We thank the reviewer for pointing out that statements could seem over-stated. While we made an effort to be as explicit, some of our excitement from addressing this novel application could have creeped into the writing. We have noted to make sure to revise results description and conclusions such that these are as explicit as possible.
>
> - Indeed, as the reviewer points out there are limitations and failure cases in our pipeline. We will add analysis of  their effect on the generated music. We have done such a study as part of ablation studies we performed earlier. These studies identified limitations in style extraction mismatches and their effect on the generated music. We will report these cases and extend the analysis to additional components. Space permitting, we will also discuss additional potential limitations and our choices in the components, such as for example, piano rolls are learned from a single music dataset - Lakh Midi dataset. The musical pieces in this dataset have certain musical styles. The model learned from this data is likely to generate patterns with such a style even for different scenarios. Given this, Drum2Music is a possible step to boost the richness of the music, which could make the video and the soundtrack more interesting, but also could generate music with repetitive style unless trained on additional datasets.
>
> - Thank you for pointing out additional negative impacts that could arise.  We will discuss this concern in the Societal Impact section. Such a concern is shared among majority of nowadays music generation systems that learn from collections of musicians or composers, and thus include the concern to be too similar to copyrighted music.

---

> > ### Comment · Reviewer_587L · 2021-08-21
> > **Reply**
> >
> > I would like to thank the authors for their extensive reply and trying to address all of my concerns. As I explained in my initial review, if all my concersns are addressed in the final version, I would be happy to increase my score. One thing that I would like to clarify was that the extra ablation study suggestion that I gave was about having a qualitative assessment comparing the proposed model and a video with a randomly sampled background rythm. As far as I understand, right now the new human assessment still contains only random rythm but also the baseline drum generation (which is not a good baseline since the annotators are asked only about synchrony). There should be a qualitative assessment that contains a subjective preference score between the proposed model and a randomly sampled (rythm + drum) from the dataset.

---

> > > ### Author Response · Authors · 2021-08-25
> > > **Reply**
> > >
> > > We thank the reviewer for clarification re. shuffled baseline subjective preference. We conducted several perceptual experiments with a general audience (mech turk) to test it.
> > >
> > > We asked the preference of in-the-wild videos with soundtracks of our generated drums vs. drums track randomly sampled from the entire dataset.
> > >
> > >  **All in-the-wild videos**
> > >
> > > |  	| Our proposed method 	| Sampled from the dataset (drum+rhythm) 	|
> > > |---	|---	|---	|
> > > | Preference 	| 55.1% (281) 	| 44.9% (229) 	|
> > > **Ratio= 1.23**
> > >
> > > Then, we looked only at a subset of these videos in which the rhythm appears perceptually to be profound (rapid movements videos).
> > >
> > > **In-the-wild videos with rapid movements**
> > >
> > > |  	| Our proposed method 	| Sampled from the dataset (drum+rhythm) 	|
> > > |---	|---	|---	|
> > > | Preference 	| 60.7% (164) 	| 39.3% (106) 	|
> > > **Ratio= 1.53**
> > >
> > > To better interpret these results, for reference, we also conducted a similar experiment with AIST dance dataset, where we tested the preference of original GT dance soundtrack vs soundtracks sampled from GT of other videos. The results are as follows:
> > >
> > > **AIST dance soundtracks**
> > >
> > > |  	| Dance soundtrack (GT full music) 	| Sampled from dataset (Shuffled GT full music) 	|
> > > |---	|---	|---	|
> > > | Preference 	| 54.9% (280) 	| 45.1% (230) 	|
> > > **Ratio= 1.22**
> > >
> > > From these experiments we arrive to the following insights:
> > > - The surveyed audience appears to prefer the soundtrack generated for the particular video over the soundtracks generated for other videos in the dataset.
> > >
> > > - When movements seem visually rapid and rhythm is distinct, the preference is stronger.
> > >
> > > - We find that a similar ratio of preference holds when we test preference of GT music vs GT sampled from the dataset.
> > >
> > > - As the reviewer points out and is supported by reference with GT shuffle, subjective preference is non unique and there could be several versions of acceptable soundtracks that match the video.
> > >
> > > - The ratio between actual soundtrack and randomly selected soundtrack of similar videos in nature turns out to be not large (1.22-1.53). This could be due to overall similarity of video clips, length of clips tested (short clips), sensitivity of the audience to music components and visual correspondence. Also, this indicates that further improvement in resolution of rhythm inference could potentially generate a better correspondence of soundtrack and movements and increase the subjective preference ratio (as we see in rapid movements videos).
> > >
> > > We will include these results and discussion in the final version of the paper.

---

> > > > ### Comment · Reviewer_587L · 2021-08-25
> > > > **Reply 2**
> > > >
> > > > Thanks a lot for running those extra subjective evaluations. If all the above hold and the authors provide the new final version addressing all reviewers' comments then I think that paper would be publication worthy.

---

### Official Review · Reviewer_oiP8 · 2021-07-15

**Rating:** 4
**Confidence:** 3

**Summary:**

The authors propose a method to generate musical soundtracks for silent videos of human activity. The method is named RhythmNet and is composed of three parts: Video2Rhythm, Rhythm2Drum, and Drum2Music. These three are sequential modules which take an input silent video and generate a musical soundtrack to accompany the video.

The Video2Rhythm component predicts musical beats given the video. It uses keypoints extracted from the video and encodes the rhythm as the beat and style. The Rhythm2Drum component uses the predicted rhythm and generates a drum pattern following the setup of GrooveVAE using a transformer-based encoder-decoder architecture followed by U-net for velocity and offset prediction. Finally, the Drum2Music component adds either piano or guitar accompaniment to the drum track to make a full musical soundtrack. Each of these individual components are independently trained.

**Limitations And Societal Impact:**

Yes, in the conclusion the authors briefly discuss this.

**Main Review:**

Positives:
- The authors indeed develop a novel approach to generate music from a given video of human motion.
- The proposed method is technically solid and the authors also present improvements over previous models for rhythm detection from video and for drum sequence generation.
- The authors use objective evaluation metrics to evaluate the various components of their method

Negatives:
- The introduction consists of various claims about the composition process in music that need citations.
- I understand the important role rhythm plays in music, but the paper makes it seem like drums are the only way to generate rhythm. This is not the case, one can simply play keys on a piano following a certain pattern and generate a musical rhythm. In fact, for videos in the wild, perhaps it is not a good idea to try to generate drums if the movements are not periodic since the drums track will simply sound out of rhythm.
- While the numbers say that the proposed method is better than previous approaches, the videos shared in the supplementary material do not sound good. The rhythm is off in almost all cases and the audio is not in sync with the video.
- While I mention earlier that the proposed method is technically solid, I’m not sure why a simple post processing of the rhythm outputs is not applied to fix some mistakes in the beat positions. Especially for the dance videos where there is a fixed tempo. Perhaps what is missing is a differentiation between beats and downbeats in the video to rhythm component.
- The paper is not well organized especially the methods section. Please add subsections and fix the spacing between paragraphs.

Overall Assessment: As mentioned, this paper demonstrates solid engineering by combining 3 independent components to approach a novel problem. While I cannot point out serious issues with the paper apart from the organization, I am not sure if the results are good enough for acceptance. Thus, the negatives slightly outweigh the positives in this paper and I rate this paper as a 4.

Comments:
- I would consider adding a subtitle in the paper title. As it is, one cannot understand what the theme of the paper is.


**Time Spent Reviewing:**

7

---

> ### Author Response · Authors · 2021-08-10
> **Reply to Reviewer oiP8**
>
> We thank the reviewer for a thoughtful review and valuable feedback. We have addressed all reviewer’s comments and concerns, in particular, regarding organization and results quality, and would like to request the reviewer to reconsider the rating.
> Below we address the items mentioned in the review.
> - We thank the reviewer for pointing out the need for additional references. We will add citations to the following references:
>   *Montagu, Jeremy, “How music and instruments began: a brief overview of the origin and entire development of music, from its earliest stages”, Frontiers in Sociology, 2017*
>
>   *Merker, Bjorn H., Guy S. Madison, and Patricia Eckerdal. "On the role and origin of isochrony in human rhythmic entrainment." Cortex 45.1 (2009): 4-17.*
>
>   *Blades, James. Percussion instruments and their history. Bold Strummer Limited, 1992.*
> in the introduction section of the final version of the paper. These support the construction of our approach.
>
> - Indeed, rhythm could be generated in various ways. We identified drums as the most direct and intuitive approach to achieve association of sounds with human movements, especially for in-the-wild videos with various movements. Additional approaches could be interesting future work extensions. In fact, surveys in which we asked people which soundtrack they prefer for in-the-wild videos, indicate that preference for only drums-only track is similar to the preference of music with drums and additional instruments. These suggest that drums are important for synchrony between audio and movements.
>   **Soundtrack Preference (Total surveyed: 850)**
> |       | No Soundtrack | Drums Only  | Drums+Piano | Drums+Guitar |
> |-------|---------------|-------------|-------------|--------------|
> | Votes | 7.3% (62)     | 31.2% (265) | 32.1% (273) | 29.4% (250)  |
>
>   Drums in our approach serve as a foundation layer in supporting the composition of the final music output. Drum tracks include additional content, e.g., snare, kick, clap and these are supposed to follow the beat and the rhythm of the movement. Noteworthy, rhythm is not necessarily periodic since beyond the beat the rhythm includes the patterns of transitional incidences, the style, (sudden changes in movement) are non-periodic signals. Rhythm is the combination of patterns of transitional movements and beats, therefore is non-periodic.
>
>
>
>
>    - Synchrony: There are of course limitations to our methodology that we will elaborate on in the final version of the paper (especially with respect to the style) but both evaluations that we consider: (i) dance videos with GT beat and (ii) in the wild videos, there is a substantial amount of synchrony and match of audio-visual content so we do not agree with the determination that the results are out of sync.
>       - Dance videos are intended for comparison with GT beat. Both numerical results and surveys reported in the paper indicate that our generated tracks are more synchronous than other approaches. We do apply the fixed tempo constraint to the generated beat and the evaluation for dance videos shows that the beats are aligned with the GT music beats.. Noteworthy, rhythm is not necessarily periodic, since while the beat is periodic, the patterns captured by transitional episodes (sudden changes in movement) are non-periodic signals. Rhythm is the combination of patterns of transitional movements and beats, and as a whole is non-periodic. For dance videos, subjective evaluation is expected to be done as an ‘overall audio-visual impression’,  e.g. we ask mechanical turkers to rate the synchrony of the whole video, vs specific attempts to follow movements or beats, since the generated soundtrack is aimed to reflect the rhythm (movements and style) of the video to provide a fuller audio-visual match. We find that both components are needed for such an experience. The style appears to be important to generate richer rhythms which correspond to movements while the beat guarantees that the overall tempo is preserved.
>        - Since our application is ultimately targeted toward videos for which there is no given background soundtrack (in-the-wild videos) the second set of examples are such videos. The survey that we report above, indicates a clear preference for a soundtrack vs silent video and an additional survey below demonstrates the importance of matching the rhythm for improving the perception of the vidoe and the soundtrack. It suggests that our methodology indeed corresponds to better rhythm prediction and music generation, which in turn leads to better soundtrack perception.
>           **Soundtrack match to the video (ablations) (Total Count: 850)**
> |       | Random Rhythm + Baseline Drum Generation (GrooVAE) | Video2Rhythm (Ours)  + Baseline Drum Generation (GrooVAE) | Video2Rhythm (Ours)  +  Rhythm2Drum (Ours) |
> |-------|----------------------------------------------------|-----------------------------------------------------------|--------------------------------------------|
> | Votes | 23.3% (198)                                        | 33.3% (283)                                               | 43.4% (369)                                |
>
>
>
> - Organization: We will follow the recommendation of the reviewer and revise the organization of the methods section and add additional explanation of the style extraction methodology. Specifically, we will split the “Music Beats Prediction” paragraph into 3 separate paragraphs describing the following topics:
>    L130-L146 - Obtaining motion features based on human body keypoints
>    L147-L164 - Performance of probabilistic beat prediction
>    L165-L169 - Discussion of the design choices for the model
>    We will also clarify the style extraction part according to the following exposition of steps:
> We define the `style’ as a combination of beat and incidences of transitional movements of the human body, such as rapid and sudden movements. Beat is a binary periodic signal determined by fixed tempo, and it is obtained by music beat prediction network, which learns the beat by pairing body keypoints and ground truth music beats in a supervised way.
> For transitional movement prediction, we apply a rule-based approach since the definition of style is implicit and there is no data to learn a mapping from body keypoints to transitional movements. We follow several key steps to obtain the transitional movements signal which is a binary signal that indicates transition timepoints as 1s and non-transitional timepoints as 0s.
> i) The first step is to compute kinematic offsets. Kinematic offsets represent the average acceleration of the human body across time, and it is a 1D time series signal.
> ii) Next we perform Short-Time-Fourier-Transform on the kinematic offsets signal to identify peaks in the change of acceleration. The highest frequency bin in STFT (out of 8) represents the most transitional parts of the signal.
> iii) The peaks are defined as 10% top magnitudes over the duration of the video. We mark the timepoints of the peaks as 1s and other timepoints as 0s.
> iv) STFT results with low temporal resolution (due to hop-size set to 4 for efficient computation) and therefore we upsample the binary signal by the hop size to obtain a binary signal that matches the resolution of the video.
>
> - We chose the title for our paper to be general and concise to intrigue the readers in the possibility of associating sound with a variety of videos with peoples’ activity. We agree that additional details would be useful. We will thereby add a clarifying subtitle:
>   **How Does it Sound? Generation of Rhythmic Soundtrack for Human Movement Videos**

---

### Official Review · Reviewer_rRNM · 2021-07-16

**Rating:** 7
**Confidence:** 4

**Summary:**

The authors propose a RhythmicNet that can generate a musical soundtrack associating with human movements in a video. Unlike most existing works that aim to generate sync sounds of audio sources in videos, this work focuses on generating musical rhythm and beat along with visual motions. It mainly consists of three parts: Video2Rhythm, Rhythm2Drum, and Drum2Music. Experiments can validate that the proposed RhythmicNet can generate plausible soundtracks for different body movements.

***Post-Rebuttal***

The rebuttal has successfully addressed my major concerns. Thus, I would like to keep my positive rating. The authors should revise the main paper by improving writing and adding new results and discussions provided in the rebuttal.

**Limitations And Societal Impact:**

Yes, the authors discussed limitations and potential ethic concerns.

**Main Review:**

Pros:

+ The rhythmic sound generation problem is interesting. Although several components, such as ST-GCN, Transformer, MIDI, and Key points have been used in previous visual-to-music sound generation methods, the proposed whole three-stage system for rhythmic sound generation is novel.

+ Extensive ablation studies are conducted for each main module. The experimental results show that the proposed modules are better than baselines.

+ The provided demo with video examples is a nice complement to demonstrate the effectiveness of the proposed method.

Cons:

- The method section is relatively hard to follow since texts with symbols and equations are densely presented (see page 4).  I would like to suggest that the authors divide the section into subsections with subsubsections to better illustrate each component in the proposed modules.

-  The proposed domain knowledge-inspired three-stage network can generate plausible sounds from body movements. However, it still requires some annotations to facilitate the learning process (e.g., Video2Rhyth), which makes it impossible to learn the model using large-scale unlabeled videos. I am wondering whether we can directly learn the video-to-drum or even video-to-music mapping without separated Video2Rhythm, Rhythm2Drum, and Drum2Music. We definitely can do it as previous methods. But, the question is whether they can achieve comparable results as the proposed method. Did the authors conduct any experiments to check results from a one-stage framework?

- Considering this is a three-stage system, it would be good if the authors can provide some failure examples for each stage and demonstrate how each stage affects final generated music sounds.

Overall, the investigated problem is interesting and the proposed method is novel and technically sound. I only have some minor questions and hope the authors can address them during rebuttal.





**Time Spent Reviewing:**

3

---

> ### Author Response · Authors · 2021-08-10
> **Reply to Reviewer rRNM**
>
> We thank the reviewer for a thoughtful review, valuable feedback and acknowledgement of our work. We address the questions raised in the review below.
> - We will follow the suggestion of the reviewer and split the music beat prediction section into subsections describing each component of our proposed model. Specifically, we will introduce three subsections:
> L130-L146 - Obtaining motion features based on human body keypoints
> L147-L164 - Performance of probabilistic beat prediction
> L165-L169 - Discussion of the design choices for the model
> and revise text accordingly.
>
> - Re. achieving similar outputs by directly learning the mapping from video to drum or even video to full music. Such general transformation is indeed desirable and could be a potential future direction. We started our work by considering such a potential model, however, we were unable to generate admissible music and thus broke the task into components. Implementation of direct approaches turned out to be surprisingly difficult and typically the output did not correspond to music at all. One of the reasons for the difficulty is due to the lack of a direct dataset that can be used to learn the mapping from movements to drum tracks with MIDI. Such a dataset is challenging to construct as well. This led us to design a componentwise generation process which utilizes available and well established datasets and methods to achieve the task.
>   It is noteworthy to mention perceptual experiments that we performed with respect to preference of people re. the kind of musical instruments to be included in the soundtrack.
>   **Soundtrack Preference (Total surveyed: 850)**
> |       | No Soundtrack | Drums Only  | Drums+Piano | Drums+Guitar |
> |-------|---------------|-------------|-------------|--------------|
> | Votes | 7.3% (62)     | 31.2% (265) | 32.1% (273) | 29.4% (250)  |
>
>    It is interesting to find that drums-only track turned out to be of similar preference as tracks with more instruments and richer music.
>
> - We will follow the suggestion of the reviewer and add analysis of limitations and failure cases per each stage and their effect on the generated music. We have done such a study as part of ablation studies we performed earlier. These studies identified limitations in style extraction mismatches and their effect on the generated music. We will report these cases and extend the analysis to additional components.

---

### Official Review · Reviewer_mrpe · 2021-07-17

**Rating:** 6
**Confidence:** 4

**Summary:**

The paper proposes an approach to generate rhythmic music from a human movement video. In detail, the model is three step process: 1) Video2Rhythm: Uses the patterns in the motion of human skeleton to detect a rhythm. 2) Rhytm2Drum: Convert the rhythmic patterns to drum beats and 3) Drum2Music: Use the drum beats to update the remi representation of another instrument (or multi-instrument song). These three steps are trained independently in a supervised learning paradigm.


**Limitations And Societal Impact:**

Yes, the paper has a discussion section explaining the limitations and potential societal impact this work could have.

**Main Review:**

## Strengths
- The paper proposes an interesting and potentially useful task of generating aligned music/beats for rhythmic human body motion
- The approach involves training three modules that can be trained separately. An advantage here is that this approach does not require a singular dataset with all the output modalities. Instead multiple and independent datasets can be used to train these three modules which can be later connected in a sequential pipeline.
- The encoder for body poses is a spatio-temporal graph convolutional network which can potentially model the hierarchy of the connections of the human body skeleton.

## Weaknesses
- A key contribution here is to use a self-similarity matrix as a prior to detect repetitions over time. A similar, and more general, approach was also proposed in [P1] which can detect repetitive patterns over time and it can do so in a self-supervised manner, unlike the proposed approach in this paper.
- Another important modelling step seems to be style extraction (l170). While the motivation to extract style seems reasonable, the effect of obtaining kinematic offsets (as a proxy for style) could benefit from more rigorous experimentation.
- The modelling choices for the Drum2Music module have not been compared with existing work such as [P2] in the paper

## Originality
Overall the methods for each separate module are well-known. The combination is somewhat unique for a task of generating music aligned with human body motion. Although the task itself is not new [P3], data-driven learning has not yet been done for this particular task.

## Quality
The paper is generally well supported with experiments except for some missing relevant comparisons in literature. See weaknesses section for a more detailed review.

## Clarity
The paper is clear and easy to understand. The only issue in terms of clarity was the section about style extraction. The definition and its importance in the pipeline weren't very clear. Maybe an example here could have made the definition more clear and an additional ablation study to study its impact would have been helpful

## Significance
While it is interesting to see a neat application of three well-studied tasks when applied together, the work presented in this paper is mostly incremental.

[P1] - Dwibedi, Debidatta, et al. "Counting out time: Class agnostic video repetition counting in the wild." Proceedings of the IEEE/CVF Conference on Computer Vision and Pattern Recognition. 2020.

[P2] - Lattner, Stefan, and Maarten Grachten. "High-level control of drum track generation using learned patterns of rhythmic interaction." 2019 IEEE Workshop on Applications of Signal Processing to Audio and Acoustics (WASPAA). IEEE, 2019.

[P3] - Berg, Tamara, et al. "Interactive music: Human motion initiated music generation using skeletal tracking by kinect." Proc. Conf. Soc. Electro-Acoustic Music United States. 2012.

**Time Spent Reviewing:**

4

---

> ### Author Response · Authors · 2021-08-10
> **Reply to Reviewer mrpe**
>
> We thank the reviewer for a thoughtful review and valuable feedback. Below we address the items mentioned in the review. Our work has similarities with [P1], [P2], [P3] to construct the pipeline of RhytmicNet, however, the details of model architecture and algorithms are substantially different as we discuss below. We will add citations to these references along with the related discussion. We discuss these references below.
> - Indeed, the self-similarity matrix in VideoToRhythm component is inspired by a similar matrix employed in [P1].  However, the outputs of VideoToRhythm and [P1] are different. In [P1] the goal is to count repetitions (mark the period) of periodic movements. In contrast, beat association corresponds to a more general task of association of a period with a signal that might be non-periodic since we consider various movements that are typically non periodic. The assumption of periodicity of movement in [P1] allows incorporation of augmentation which leads to successful self-supervised learning. In our case, such augmentation and inclusion of synthetic data in a similar manner is not practical. We find that a fully supervised approach is needed to successfully learn the transformation.
> - In [P2] a model that generates kick-drum music given conditional signals such as beat, downbeat, onset of Snare and Bass is introduced. The output is a time-series of 1D continuous signal of kick track. In contrast, Drum2Music's objective is different from the objective of [P2]. It generates a polyphonic MIDI track such as guitar/piano conditioned on a Drum MIDI,  i.e. generates music of additional instruments on top of the drums track MIDI. The MIDI content consists of not only binary onsets but also velocity and time offsets.
>
> - [P3] is a classical rule-based approach for conversion of sensor system detected motion to music notes. TThe algorithm is based on frequency modulation that takes human body movements as input.The rule is governed by frequency modulation that takes human body movements as input. This work shows that it is possible to correlate movement and sounds. Our pipeline is a learning-based approach that aims to achieve a detailed and synchronic music soundtrack with movements.
>
> - Style extraction is an important step to obtain a detailed and richer rhythm. This step is rule-based algorithm (vs learned-based). Our approach detects transitional incidences reasonably well. We intend to illustrate the style extraction algorithm more clearly in the final version following the following explanation:
>    - We define the 'style' as a combination of beat and incidences of transitional movements of the human body, such as rapid and sudden movements. Beat is a binary periodic signal determined by fixed tempo, and it is obtained by music beat prediction network, which learns the beat by pairing body keypoints and ground truth music beats in a supervised way.
>    - For transitional movement prediction, we apply a **rule-based** approach since the definition of style is implicit and there is no data to learn a mapping from body keypoints to transitional movements. We follow several key steps to obtain the transitional movements signal which is a binary signal that indicates transition timepoints as 1s and non-transitional timepoints as 0s.
>      i) The first step is to compute **kinematic offsets**. Kinematic offsets represent the average acceleration of the human body across time, and it is a 1D time series signal.
>      ii) Next we perform **Short-Time-Fourier-Transform** on the kinematic offsets signal to identify peaks in the change of acceleration. The highest frequency bin in STFT (out of 8) represents the most transitional parts of the signal.
>      iii) The peaks are defined as 10% top magnitudes over the duration of the video. We mark the timepoints of the peaks as 1s and other timepoints as 0s.
>      iv) STFT results with low temporal resolution (due to hop-size set to 4 for efficient computation) and therefore we upsample the binary signal by the hop size to obtain a binary signal that matches the resolution of the video.

---

> > ### Comment · Reviewer_mrpe · 2021-08-31
> > **Response to Authors**
> >
> > I thank the authors for their detailed response. The authors have answered my questions for the most part.
> > Based on the other reviews (+responses), it seems that style extraction still remains a point of discussion. My main concern with this paper is also the style extraction of the rhythm. While the definition of style has been limited to only beat and transitional movements of the human body, which is okay as long as it provides the correct inductive biases for the application, the experiments are not thorough enough to judge what was the contribution of either of the *Music Beat Prediction* or the *Style Extraction* component of the Video2Rhythm step. It is possible that *Style Extraction* dominates the prediction of the rhythm. Either way, the impact of each of the submodules would be useful information for the readers. Specifically, an ablation study where either style or beats are omitted would be useful.
> >
> > Another related concern is that the definition of Style appears to be a superset of the definition of Music Beats.
> > > We define the 'style' as a combination of beat and incidences of transitional movements of the human body, such as rapid and sudden movements.
> >
> > Currently, it appears that the rule-based approach (i.e. beats in Style Extraction) and learned approach (i.e. Music Beat Prediction) are being fused by addition (or concatenation- was not completely clear from in the paper). This is probably fine, but makes it even important to study the impact of the two submodules.

---

> > > ### Author Response · Authors · 2021-09-02
> > > **Reply to Reviewer mrpe**
> > >
> > > We thank the reviewer for reiterating the meaning of style from our first reply. We realized that style and rhythm in the first sentence were switched. Instead of “We define the 'style' as a combination of beat and incidences of transitional movements of the human body, such as rapid and sudden movements.” The correct sentence is: “We define the rhythm as a combination of (i) beat and (ii) incidences of transitional movements of the human body (style) such as rapid and sudden movements.”
> > >
> > > Early in the design and the experimentation of RhythmicNet we clearly observed that associating a rhythm for generation of a soundtrack for body movements requires both components, i.e. both the beat and the style. This is expected, since the beat signal is repetitive while the style is rapid and non-repetitive. Together they complement each other and form the rhythm of the movement. Since the components are complementary, they do not compete with each other and provide a better match between the music and the movements. We will include illustrative examples in the supplementary materials of several videos for which the soundtrack is generated from the beat alone, or style alone, or the full rhythm. These clearly demonstrate the differences between the signals and the effect of combining them into a rhythm and corresponding music.

---

### Official Review · Reviewer_vA5F · 2021-07-22

**Rating:** 7
**Confidence:** 3

**Summary:**

In this paper, the authors propose to generate music based on human motion videos. The authors propose the Video2Rhythm to capture the  rhythmic nature of free body movements. Then they use Rhythm2Drum module to generate drum. Finally, piano and guitar tracks are added to enrich the music through Drum2Music. Experiments show that they can generate plausible music that aligns with the videos.

**Limitations And Societal Impact:**

The authors have addressed the negative impacts, however, the authors could include more limitations. For example, some failure cases can be shown for better illustration.

The authors claim that the code and models are proprietary, which would cast great difficult for follow-up works. As many parts are not clearly written, it might be difficult for others to reproduce the work.

**Main Review:**

Overall, I am fond of the topic and the results are indeed plausible. Though the whole paper can basically be decomposed into three independent parts with building blocks very similar with previously proposed methods such as [1,2,3]. The three modules are developed reasonably, and the design of the whole pipeline is of enough novelty for an application paper. The strengths are explicitly listed as follows:

++ The topic is interesting and worth exploring.

++ The supp video shows the effectiveness of the proposed method.

++ The formulation and the pipeline of this paper are reasonable.

++ The usage of three different datasets is clever.

The weaknesses are as follows:

1. The authors could pay more attention to the writings. The descriptions in the method part are not clear enough. The captions are not detailed enough. The authors should try replacing the current png/jpg figure format with pdf.

a) L35 “but even beforehand”, the but is strange in this sentence.

b) It would be better if the authors could re-split the “Music Beats Prediction” paragraph (starting from L130).

c) The notion of “directogram” is proposed in [1], but is not mentioned.

d) The representation of the “style” has never been clearly illustrated. The caption of figure 3 is too simple for a clear illustration.

In line 184: “We choose frames with the top 10% frequency magnitude as the style pattern to music beats for every second” what do you mean by “to music beats”?

My understanding is that for beats prediction, a binary prediction is given related to frames periodically. As for the style, according to the previous sentence, it seems also to be time steps associated with frames with sudden movements. I am not sure about “the output signal is re-sampled to have the same sampling rate as the music beats”. How to perform the re-sample? How are style and beat combined together?

e) How is the training of the rhythm part supervised? Is there any supervision for the style pattern?

2. The title of this paper is not informative enough. It somehow reminds me of “Objects that sound” or “Visually Indicated Sounds”, which correlate with general sounding objects. The authors are encouraged to change it to a one more related to motion/dance and music.

3. All building blocks are designed with sophisticated models. I am not suggesting that the authors should do more meaningless ablations, but there are obviously easier solutions. The authors are encouraged to propose certain baselines for replacing, maybe the transformer structures.

I will consider raising the score if the method part could be modified clearer with more details.

[1] Visual rhythm and beat. TOG 2018.

[2] Learning to groove with inverse sequence transformations. ICML, 2019.

[3] Transformer-XL: Attentive Language Models Beyond a Fixed-Length Context. ACL 2019.

---------------------------------------------------------------------------------------------------------------------------------------------------------------

Please make sure to include the clarifications and experiments in all responses to the final version.



**Time Spent Reviewing:**

24 hours

---

> ### Author Response · Authors · 2021-08-10
> **Reply to Reviewer vA5F**
>
> We thank the reviewer for a thoughtful review and valuable feedback. We provide our point by point clarifications and intended revisions below. In particular, in light of comment 1) we will extensively clarify the exposition of the methods in the final version.
>
> 1. We will clarify the exposition of the methodology in the final version of the methods section, following the feedback by the reviewer. We will also make overall writing improvements.
>      a). **We will change** “Indeed, drums are known to have existed from around 6000 BC, but even beforehand there were instruments based on principle of hitting two objects and generating sounds.” **to** “Indeed, drums are known to have existed from 6000 BC, and even beforehand, there were instruments based on the principle of hitting two objects and generating sounds.”
>      b).  As recommended, **we will split** the “Music Beats Prediction” paragraph into 3 separate paragraphs describing the following topics:
>      L130-L146 - Obtaining motion features based on human body keypoints
>      L147-L164 - Performance of probabilistic beat prediction
>      L165-L169 - Discussion of the design choices for the model
>     c). We cite [1] as Ref. [48] in the context of kinematic offsets, i.e., L174-175: “We infer changing patterns of motion strength by kinematic offsets [47]&[48].” We **will move the citation** of Ref. [48] to the location where the directogram is introduced.
>     d). We agree with the reviewer that the style extraction part could be illustrated more clearly. In Figure 3, its caption and main text, we will include a more detailed explanation of style extraction.
>     - We define the 'style' as a combination of beat and incidences of transitional movements of the human body, such as rapid and sudden movements. Beat is a binary periodic signal determined by fixed tempo, and it is obtained by music beat prediction network, which learns the beat by pairing body keypoints and ground truth music beats in a supervised way.
>     - For transitional movement prediction, we apply a **rule-based** approach since the definition of style is implicit and there is no data to learn a mapping from body keypoints to transitional movements. We follow several key steps to obtain the transitional movements signal which is a binary signal that indicates transition timepoints as 1s and non-transitional timepoints as 0s.
>          i) The first step is to compute **kinematic offsets**. Kinematic offsets represent the average acceleration of the human body across time, and it is a 1D time series signal.
>          ii) Next we perform **Short-Time-Fourier-Transform** on the kinematic offsets signal to identify peaks in the change of acceleration. The highest frequency bin in STFT (out of 8) represents the most transitional parts of the signal.
>          iii) The peaks are defined as 10% top magnitudes over the duration of the video. We mark the timepoints of the peaks as 1s and other timepoints as 0s.
>          iv) STFT results with low temporal resolution (due to hop-size set to 4 for efficient computation) and therefore we upsample the binary signal by the hop size to obtain a binary signal that matches the resolution of the video.
>
>     e). Identification of patterns of motion change (style part of the rhythm) is implemented with a rule-based approach rather than learning-based as we explain above. The prediction of beats (fundamental part of the rhythm) is learned in a supervised way.
>
> 2. We chose the title to be general and concise to intrigue the readers in the possibility to associate sound with a variety of videos with people activity. We agree that additional details would be useful and, thereby, we will add a clarifying subtitle:
>     **How Does it Sound? Generation of Rhythmic Soundtrack for Human Movement Videos**
>
> 3. Our model architecture is a result of multiple experimentation and investigation. We implemented basic baselines and then progressively added or changed components to make the system more robust. Our ablation studies indicated that there is a significant improvement when transformer architecture is used vs. more simpler network models. This results also in significantly better output quality, e.g. beat prediction accuracy largely affects the synchronization between audio and visual content. The benefit of our components is also seen in perceptual ablation studies that we performed show in the Table below.
>      **Soundtrack match to the video (ablations) (Total Count: 850)**
> |       | Random Rhythm + Baseline Drum Generation (GrooVAE) | Video2Rhythm (Ours)  + Baseline Drum Generation (GrooVAE) | Video2Rhythm (Ours)  +  Rhythm2Drum (Ours) |
> |-------|----------------------------------------------------|-----------------------------------------------------------|--------------------------------------------|
> | Votes | 23.3% (198)                                        | 33.3% (283)                                               | 43.4% (369)                                |
>
>
> - We will follow the suggestion of the reviewer and add analysis of limitations and failure cases and their effect on the generated music. We have done such a study as part of ablation studies we performed earlier. These studies identified limitations in style extraction mismatches and their effect on the generated music. We will report these cases and extend the analysis to additional components.
>
> - We intend to release **the code and step-by-step manual** upon the publication of our work.

---

> > ### Comment · Reviewer_vA5F · 2021-08-12
> > **Reply to Authors**
> >
> > I have read all reviews and the rebuttal of the authors. The major problems raised are shared across multiple reviewers, including:
> >
> > * The paper is not well-organized, particularly for the method part.
> >
> > * The style extraction part is not clear enough.
> >
> > * The title of this paper is not informative enough.
> >
> > * Missing ablation studies with certain baselines.
> >
> > * More analysis on failure cases.
> >
> > These problems have mostly been addressed in the rebuttal. If the authors ensure that these modifications as well as the code and models can be updated to the final version, then I think this paper is worth accpetance. It indeed brings a solution to an interesting problem.

---

### Decision · Program_Chairs · 2021-09-27

**Decision:**

Accept (Poster)

**Comment:**

This paper received 4 positive reviews and 1 negative review. In the rebuttal, the authors have addressed most of the concerns. AC feels this work is very interesting and deserves to be published on NeurIPS 2021. The reviewers did raise some valuable concerns (e.g., including more human studies) that should be addressed in the final camera-ready version of the paper. The authors are encouraged to make other necessary changes.